# DOES INJECTING LINGUISTIC STRUCTURE INTO LANGUAGE MODELS LEAD TO BETTER ALIGNMENT WITH BRAIN RECORDINGS?

## ABSTRACT

Neuroscientists evaluate deep neural networks for natural language processing as possible candidate models for how language is processed in the brain. These models are often trained without explicit linguistic supervision, but have been shown to learn some linguistic structure in the absence of such supervision (Manning et al., 2020), potentially questioning the relevance of symbolic linguistic theories in modeling such cognitive processes (Warstadt & Bowman, 2020). We evaluate across two fMRI datasets whether language models align better with brain recordings, if their attention is biased by annotations from syntactic or semantic formalisms. Using structure from dependency or minimal recursion semantic annotations, we find alignments improve significantly for one of the datasets. For another dataset, we see more mixed results. We present an extensive analysis of these results. Our proposed approach enables the evaluation of more targeted hypotheses about the composition of meaning in the brain, expanding the range of possible scientific inferences a neuroscientist could make, and opens up new opportunities for cross-pollination between computational neuroscience and linguistics.

## 1 INTRODUCTION

Recent advances in deep neural networks for natural language processing (NLP) have generated excitement among computational neuroscientists, who aim to model how the brain processes language. These models are argued to better capture the complexity of natural language semantics than previous computational models, and are thought to represent meaning in a way that is more similar to how it is hypothesized to be represented in the human brain. For neuroscientists, these models provide possible hypotheses for *how* word meanings compose in the brain. Previous work has evaluated the plausibility of such candidate models by testing how well representations of text extracted from these models align with brain recordings of humans during language comprehension tasks (Wehbe et al., 2014; Jain & Huth, 2018; Gauthier & Ivanova, 2018; Gauthier & Levy, 2019; Abnar et al., 2019; Toneva & Wehbe, 2019; Schrimpf et al., 2020; Caucheteux & King, 2020), and found some correspondences.

However, modern NLP models are often trained without explicit linguistic supervision (Devlin et al., 2018; Radford et al., 2019), and the observation that they nevertheless learn some linguistic structure has been used to question the relevance of symbolic linguistic theories. Whether injecting such symbolic structures into language models would lead to even better alignment with cognitive measurements, however, has not been studied. In this work, we address this gap by training BERT (§3.1) with structural bias, and evaluate its alignment with brain recordings (§3.2). Structure is derived from three formalisms—UD, DM and UCCA (§3.3)—which come from different linguistic traditions, and capture different aspects of syntax and semantics.

Our approach, illustrated in Figure 1, allows for quantifying the brain alignment of the structurally-biased NLP models in comparison to the base models, as related to new information about linguistic structure learned by the models that is also potentially relevant to language comprehension in the brain. More specifically, in this paper, we:

(a) Employ a fine-tuning method utilising structurally guided attention for injecting structural bias into language model (LM) representations.

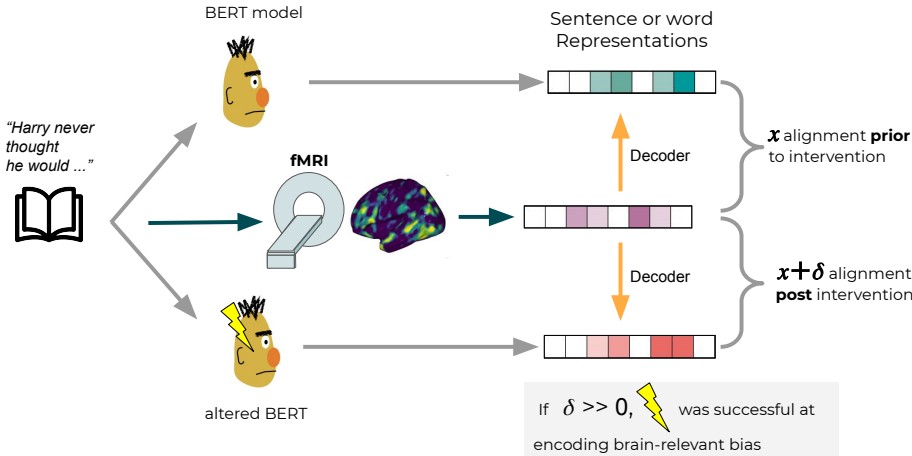

Figure 1: Overview of our approach. We use BERT as a baseline and inject structural bias in two ways. Through a brain decoding task, we then compare the alignment of the (sentence and word) representations of our baseline and our altered models with brain activations.

(b) Assess the representational alignment to brain activity measurements of the fine-tuned and non-fine-tuned LMs.

(c) Further evaluate the LMs on a range of targeted syntactic probing tasks and a semantic tagging task, which allow us to uncover fine-grained information about their structure-sensitive linguistic capabilities.

(d) Present an analysis of various linguistic factors that may lead to improved or deteriorated brain alignment.

## 2 BACKGROUND: BRAIN ACTIVITY AND NLP

Mitchell et al. (2008) first showed that there is a relationship between the co-occurrence patterns of words in text and brain activation for processing the semantics of words. Specifically, they showed that a computational model trained on co-occurrence patterns for a few verbs was able to predict fMRI activations for novel nouns. Since this paper was introduced, many works have attempted to isolate other features that enable prediction and interpretation of brain activity (Frank et al., 2015; Brennan et al., 2016; Lopopolo et al., 2017; Anderson et al., 2017; Pereira et al., 2018; Wang et al., 2020). Gauthier & Ivanova (2018) however, emphasize that directly optimizing for the decoding of neural representation is limiting, as it does not allow for the uncovering of the mechanisms that underlie these representations. The authors suggest that in order for us to better understand linguistic processing in the brain, we should also aim to train models that optimize for a specific linguistic task and explicitly test these against brain activity.

Following this line of work, Toneva & Wehbe (2019) present experiments both predicting brain activity and evaluating representations on a set of linguistic tasks. They first show that using uniform attention in early layers of BERT (Devlin et al., 2018) instead of pretrained attention leads to better prediction of brain activity. They then use the representations of this altered model to make predictions on a range of syntactic probe tasks, which isolate different syntactic phenomena (Marvin & Linzen, 2019), finding improvements against the pretrained BERT attention. Gauthier & Levy (2019) present a series of experiments in which they fine-tune BERT on a variety of tasks including language modeling as well as some custom tasks such as scrambled language modeling and part-of-speech-language modeling. They then perform brain decoding, where a linear mapping is learnt from fMRI recordings to the fine-tuned BERT model activations. They find that the best mapping is obtained with the scrambled language modelling fine-tuning. Further analysis using a structural probe method

confirmed that the token representations from the scrambled language model performed poorly when used for reconstructing Universal Dependencies (UD; Nivre et al., 2016) parse trees.

When dealing with brain activity, many confounds may lead to seemingly divergent findings, such as the size of fMRI data, the temporal resolution of fMRI, the low signal-to-noise ratio, as well as how the tasks were presented to the subjects, among many other factors. For this reason, it is essential to take sound measures for reporting results, such as cross-validating models, evaluating on unseen test sets, and conducting a thorough statistical analysis.

## 3 APPROACH

Figure 1 shows a high-level outline of our experimental design, which aims to establish whether injecting structure derived from a variety of syntacto-semantic formalisms into neural language model representations can lead to better correspondence with human brain activation data. We utilize fMRI recordings of human subjects reading a set of texts. Representations of these texts are then derived from the activations of the language models. Following Gauthier & Levy (2019), we obtain LM representations from BERT[1] for all our experiments. We apply masked language model fine-tuning with attention guided by the formalisms to incorporate structural bias into BERT's hidden-state representations. Finally, to compute alignment between the BERT-derived representations—with and without structural bias—and the fMRI recordings, we employ the brain decoding framework, where a linear decoder is trained to predict the LM derived representation of a word or a sentence from the corresponding fMRI recordings.

### 3.1 LM-DERIVED REPRESENTATIONS

BERT uses wordpiece tokenization, dividing the text to sub-word units. For a sentence $S$ made up of $P$ wordpieces , we perform mean-pooling over BERT's final layer hidden-states $[h_1, ..., h_P]$, obtaining a vector representation of the sentence $S_{mean} = \frac{1}{P} \sum_p h_p$ (Wu et al., 2016). In initial experiments, we found that this leads to a closer match with brain activity measurements compared to both max-pooling and the special `[CLS]` token, which is used by Gauthier & Levy (2019). Similarly, for a word $W$ made up of $P$ wordpieces, to derive word representations, we apply mean-pooling over hidden-states $[h_1, ..., h_P]$, which correspond to the wordpieces that make up $W$: $W_{mean} = \frac{1}{P} \sum_p h_p$. For each dataset, $D_{LM} \in \mathbb{R}^{n \times d_H}$ denotes a matrix of $n$ LM-derived word or sentence representations where $d_H$ is BERT's hidden layer dimensionality ($d_H = 1024$ in our experiments).

### 3.2 NEUROIMAGING DATASETS

We utilize two fMRI datasets, which differ in the granularity of linguistic cues to which human responses were recorded. The first, collected in Pereira et al. (2018)'s experiment 2, comprises a single brain image per entire sentence. In the second, more fine-grained dataset, recorded by Wehbe et al. (2014), each brain image corresponds to 4 words. We conduct a **sentence-level** analysis for the former and a **word-level** one for the latter.[2]

**Pereira2018** consists of fMRI recordings from 8 subjects. The subjects were presented with stimuli consisting of 96 Wikipedia-style passages written by the authors, consisting of 4 sentences each. The subjects read the sentences one by one and were instructed to think about their meaning. The resulting data for each subject consists of 384 vectors of dimension 200,000; a vector per sentence. These were reduced to 256 dimensions using PCA by Gauthier & Levy (2019). These PCA projections explain more than 95% of the variance among sentence responses within each subject. We use this reduced version in our experiments.

**Wehbe2014** consists of fMRI recordings from 8 subjects as they read a chapter from *Harry Potter and the Sorcerer's Stone*. For the 5000 word chapter, subjects were presented with words one by one for 0.5 seconds each. An fMRI image was taken every 2 seconds, as a result, each image corresponds

---

[1]Specifically: `bert-large-uncased` trained with whole-word masking.

[2]Even though the images are recorded at the 4-gram level of granularity, a word-level analysis is applied, as in Schwartz et al. (2019).

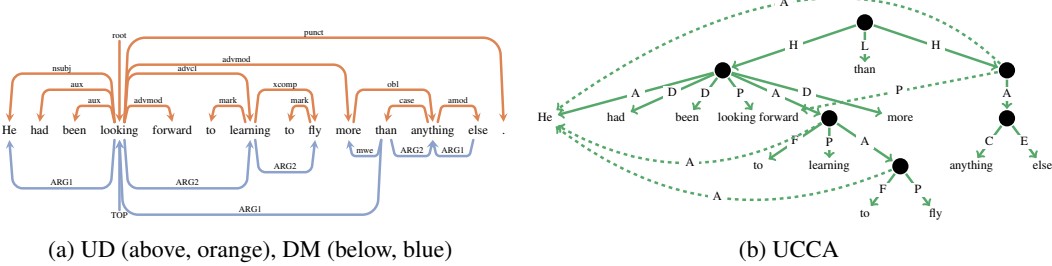

(a) UD (above, orange), DM (below, blue)  (b) UCCA

Figure 2: Manually annotated example graphs for a sentence from the Wehbe2014 dataset. While UCCA and UD attach all words, DM only connects content words. However, all formalisms capture basic predicate-argument structure, for example, denoting that "more than anything else" modifies "looking forward" rather than "fly".

to 4 words. The data was further preprocessed (i.e. detrended, smoothed, trimmed) and released by Toneva & Wehbe (2019). We use this preprocessed version to conduct word-level analysis, for which we use PCA to reduce the dimensions of the fMRI images from 25,000 to 750, explaining at least 95% variance for each participant.

## 3.3 FORMALISMS AND DATA

To inject linguistic structure into language models, we experiment with three distinct formalisms for representation of syntactic/semantic structure, coming from different linguistic traditions and capturing different aspects of linguistic signal: UD, DM and UCCA. An example graph for each formalism is shown in Figure 2. Although there are other important linguistic structured formalisms, including meaning representations such as AMR (Banarescu et al., 2013), DRS (Kamp & Reyle, 1993; Bos et al., 2017) and FGD (Sgall et al., 1986; Hajic et al., 2012), we select three relatively different formalisms as a somewhat representative sample. All three have manually annotated datasets, which we use for our experiments.

UD (Universal Dependencies; Nivre et al., 2020) is a syntactic bi-lexical dependency framework (dependencies are denoted as arcs between words, with one word being the *head* and another the *dependent*), which represents grammatical relations according to a coarse cross-lingual scheme. For UD data, we use the English Web Treebank corpus (EWT; Silveira et al., 2014), which contains 254,830 words and 16,622 sentences, taken from five genres of web media: weblogs, newsgroups, emails, reviews, and Yahoo! answers.

DM (DELPH-IN MRS Bi-Lexical Dependencies; Ivanova et al., 2012) is derived from the underspecified logical forms computed by the English Resource Grammar (Flickinger et al., 2017; Copestake et al., 2005), and is one of the frameworks targeted by the Semantic Dependency Parsing SemEval Shared Tasks (SDP; Oepen et al., 2014; 2015). We use the English SDP data for DM (Oepen et al., 2016), annotated on newspaper text from the Wall Street Journal (WSJ), containing 802,717 words and 35,656 sentences.

UCCA (Universal Cognitive Conceptual Annotation; Abend & Rappoport, 2013) is based on cognitive linguistic and typological theories, primarily Basic Linguistic Theory (Dixon, 2010/2012). We use UCCA annotations over web reviews text from the English Web Treebank, and from English Wikipedia articles on celebrities. In total, they contain 138,268 words and 6,572 sentences. For uniformity with the other formalisms, we use bi-lexical approximation to convert UCCA graphs, which have a hierarchical constituency-like structure, to bi-lexical graphs with edges between words. This conversion keeps about 91% of the information (Hershcovich et al., 2017).

## 3.4 INJECTING STRUCTURAL BIAS INTO LMS

Recent work has explored ways of modifying attention in order to incorporate structure into neural models (Chen et al., 2016; Strubell et al., 2018; Strubell & McCallum, 2018; Zhang et al., 2019;

Bugliarello & Okazaki, 2019). For instance, Strubell et al. (2018) incorporate syntactic information by training one attention head to attend to syntactic heads, and find that this leads to improvements in Semantic Role Labeling (SRL). Drawing on these approaches, we modify the BERT Masked Language Model (MLM) objective with an additional structural attention constraint. BERT$_{\text{LARGE}}$ consists of 24 layers and 16 attention heads. Each attention head $head_i$ takes in as input a sequence of representations $h = [h_1, ..., h_P]$ corresponding to the $P$ wordpieces in the input sequence. Each representation in $h_p$ is transformed into query, key, and value vectors. The scaled dot product is computed between the query and all keys and a softmax function is applied to obtain the attention weights. The output of $head_i$ is a matrix $O_i$, corresponding to the weighted sum of the value vectors.

For each formalism and its corresponding corpus, we extract an adjacency matrix from each sentence's parse. For the sequence $S$, the adjacency matrix $A_S$ is a matrix of size $P \times P$, where the columns correspond to the heads in the parse tree and the rows correspond to the dependents. The matrix elements denote which tokens are connected in the parse tree, taking into account BERT's wordpiece tokenization. Edge directionality is not considered. We modify BERT to accept as input a matrix $A_S$ as well as $S$; maintaining the original MLM objective. For each attention head $head_i$, we compute the binary cross-entropy loss between $O_i$ and $A_S$ and add that to our total loss, potentially down-weighted by a factor of $\alpha$ (a hyperparameter). BERT's default MLM fine-tuning hyperparameters are employed and $\alpha$ is set to 0.1 based on validation set perplexity scores in initial experiments.

Structural information can be injected into BERT in many ways, in many heads, across many layers. Because the appropriate level and extent of supervision is unknown a priori, we run various fine-tuninig settings with respect to combinations of number of layers $(1, \dots, 24)$ and attention heads $(1, 3, 5, 7, 9, 11, 12)$ supervised via attention guidance. Layers are excluded from the bottom up (e.g.: when 10 layers are supervised, it is the topmost 10); heads are chosen according to their indices (which are arbitrary). This results in a total of 168 fine-tuning settings per formalism. For each fine-tuning setting, we perform two fine-tuning runs.[3] For each run $r$ of each fine-tuning setting $f$, we derive a set of sentence or word representations $D_{fr} \in \mathbb{R}^{n \times d_H}$ from each fine-tuned model using the approach described in §3.1 for obtaining $D_{LM}$, the baseline set of representations from BERT before fine-tuning. We then use development set[4] embedding space hubness—an indicator of the degree of difficulty of indexing and analysing data (Houle, 2015) which has been used to evaluate embedding space quality (Dinu et al., 2014)—as an unsupervised selection criterion for the fine-tuned models, selecting the model with the lowest degree of hubness (per formalism) according to the Robin Hood Index (Feldbauer et al., 2018). This yields three models for each of the two datasets—one per formalism—for which we present results below.

In addition to the approach described above, we also experiment with directly optimizing for the prediction of the formalism graphs (i.e., parsing) as a way of encoding structural information in LM representations. We find that this leads to a consistent decline in alignment of the LMs' representations to brain recordings. Further details can be found in Appendix A.

### 3.5 BRAIN DECODING

To measure the alignment of the different LM-derived representations to the brain activity measurements, brain decoding is performed, following the setup described in Gauthier & Levy (2019).[5] For each subject $i$'s fMRI images corresponding to a set of $n$ sentences or words, a ridge regression model is trained to linearly map from brain activity $B_i \in \mathbb{R}^{n \times d_B}$ ($n = 384$; $d_B = 256$ for **Pereira2018** and $n = 4369$; $d_B = 750$ for **Wehbe2014**) to a LM-derived representation ($D_{fr}$ or $D_{LM}$), minimizing the following loss:

$$\mathcal{L}_{ifr} = \|B_i G_{i \to fr} - D_{fr}\|_2^2 + \lambda \|G_{i \to fr}\|_2^2$$

where $G_{i \to fr} : \mathbb{R}^{d_H \times d_B}$ is a linear map, and $\lambda$ is a hyperparameter for ridge regularization. Nested 12-fold cross-validation (Cawley & Talbot, 2010) is used for selection of $\lambda$, training and evaluation.

---

[3] We find that the mean difference in brain decoding score (Pearson's $r$) between two runs of the same setting (across all settings) is low (0.003), indicating that random initialization does not play a major part in our results. We, therefore, do not carry out more runs.

[4] For **Wehbe2014**: second chapter of Harry Potter. For **Pereira2018**: first 500 sentences of English Wikipedia.

[5] Other methods for evaluating representational correspondence such as Representational Similarity Analysis (Kriegeskorte et al., 2008) and the Centered Kernel Alignment similarity index (Kornblith et al., 2019) were also explored but were found to be either less powerful or less consistent across subjects and datasets.

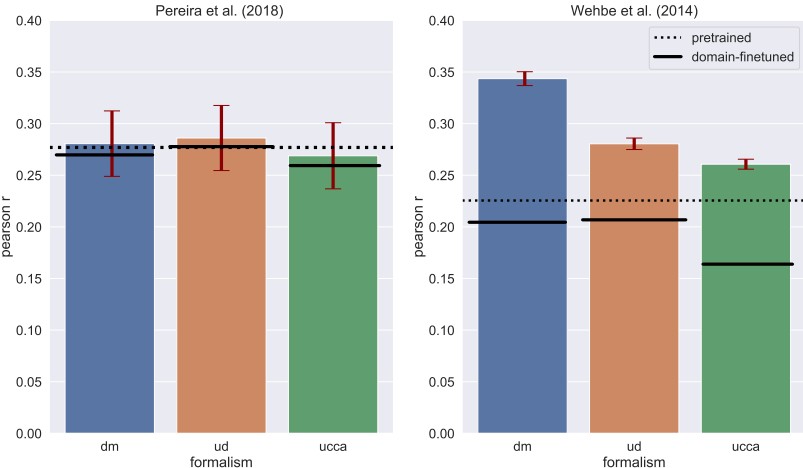

Figure 3: Brain decoding score (mean Pearson's $r$; with 95% confidence intervals shown for subject scores) for models fine-tuned by MLM with **guided attention** on each of the formalisms, as well the baseline models: pretrained BERT (dotted line), and BERT fine-tuned by MLM on each formalism's training text without guided attention (domain-finetuned BERT, solid lines).

**Evaluation**    To evaluate the regression models, Pearson's correlation coefficient between the predicted and the corresponding heldout true sentence or word representations is computed. We find that this metric[6] is consistent across subjects and across the two datasets. We run 5000 bootstrap resampling iterations and a) report the mean [7] correlation coefficient (referred to as *brain decoding score/performance*), b) use a paired bootstrap test to establish whether two models' mean (across stimuli) scores were drawn from populations having the same distribution [8], c) apply the Wilcoxon signed rank test (Wilcoxon, 1992) to the by-subject scores to for evidence of strength of generalization over subjects. Bonferroni correction (correcting for 3 multiple comparisons) is used to adjust for multiple hypothesis testing. See Appendix C for details.

## 4    RESULTS

To evaluate the effect of the structurally-guided attention, we compute the brain decoding scores for the guided-attention models corresponding to each formalism and fMRI dataset and compare these scores against the brain decoding scores from two baseline models: 1) a *domain-finetuned* BERT (DF), which finetunes BERT using the regular MLM objective on the text of each formalism's training data, and a *pretrained* BERT. We introduce the *domain-finetuned* baseline in order to control for any effect that finetuning using a specific text domain may have on the model representations. Comparing against this baseline allows us to better isolate the effect of injecting the structural bias from the possible effect of simply fine-tuning on the text domain. We further compare to a pretrained baseline in order to evaluate how the structurally-guided attention approach performs against an off-the-shelf model that is commonly used in brain-alignment experiments.

### 4.1    PEREIRA2018

Figure 3 shows the sentence-level brain decoding performance on the **Pereira2018** dataset, for the guided attention fine-tuned models (GA) and both baseline models (domain-finetuned and pretrained). We find that the domain-finetuned baseline (shown in Figure 3 as solid lines) leads to brain decoding

---

[6]Appendix B shows results for the rank-based metric reported in Gauthier & Levy (2019), which we find to strongly correspond to Pearson's correlation. This metric evaluates representations based on their support for contrasts between sentences/words which are relevant to the brain recordings. Other metrics for the evaluation of goodness of fit were found to be less consistent.

[7]Across fine-tuning runs, cross-validation splits, and bootstrap iterations.

[8]This is applied per subject to test for strength of evidence of generalization over sentence stimuli.

scores that are either lower than or not significantly different from the pretrained baseline. Specifically, for DM and UCCA, the DF baseline performs below the pretrained baseline, which suggests that simply fine-tuning on these corpora results in BERT's representations becoming less aligned with the brain activation measurements from **Pereira2018**. We find that all GA models outperform their respective DF baselines (for all subjects, $p < 0.05$). We further find that compared to the pretrained baselines, with $p < 0.05$: a) the UD GA model shows significantly better brain decoding scores for 7 out of 8 subjects, b) the DM GA model for 4 out of 8 subjects, c) the UCCA GA shows scores not significantly different from or lower, for all subjects. For details see Appendix C.

## 4.2 WEHBE2014

For **Wehbe2014**, where analysis is conducted on the word level, we again find that domain-finetuned models—especially the one finetuned on the UCCA domain text—achieve considerably lower brain decoding scores than the pretrained model, as shown in Figure 3. Furthermore, the guided-attention models for all three formalisms outperform both baselines by a large, significant margin (after Bonferroni correction, $p < 0.0001$).

## 5 DISCUSSION AND ANALYSIS

Overall, our results show that structural bias from syntacto-semantic formalisms can improve the ability of a linear decoder to map the BERT representations of stimuli sentences to their brain recordings. This improvement is especially clear for **Wehbe 2014**, where token representations and not aggregated sentence representations (as in **Pereira 2018**) are decoded, indicating that finer-grain recordings and analyses might be necessary for modelling the correlates of linguistic structure in brain imaging data. To arrive at a better understanding of the effect of the structural bias and its relationship to brain alignment, in what follows, we present an analysis of the various factors which affect and interact with this relationship.

**The effect of domain**    Our results suggest that the domain of fine-tuning data and of stimuli might play a significant role, despite having been previously overlooked: simply fine-tuning on data from different domains leads to varying degrees of alignment to brain data. To quantify this effect, we compute the average word perplexity of the stimuli from both fMRI datasets for the pretrained and DF baselines on each of the three domain datasets.[9] If the domain of the corpora used for fine-tuning influences our results as hypothesized, we expect this score to be higher for the DF baselines. We find that this is indeed the case and that for those baselines (DF), increase in perplexity roughly corresponds to lower brain decoding scores—see detailed results in Appendix D. This finding calls to attention the necessity of accounting for domain match in work utilizing cognitive measurements and emphasizes the importance of the domain-finetuned baseline in this study.

**Targeted syntactic evaluation**    We evaluate all models on a range of syntactic probing tasks proposed by Marvin & Linzen (2019).[10] This dataset tests the ability of models to distinguish minimal pairs of grammatical and ungrammatical sentences across a range of syntactic phenomena. Figure 4 shows the results for the three **Wehbe2014** models across all subject-verb agreement (SVA) tasks.[11] We observe that after attention-guided fine-tuning: a) the DM guided-attention model, and to a lesser extent the UD guided-attention model have a higher score than the pretrained baseline and the domain-finetuned baselines for most SVA tasks and b) the ranking of the models corresponds to their ranking on the brain decoding task (DM > UD > UCCA).[12] Although all three formalisms annotate the subject-verb-object or predicate-argument structure necessary for solving SVA tasks, it appears that some of them do so more effectively, at least when encoded into a LM by GA.

---

[9]Note that this is not equivalent to the commonly utilised sequence perplexity (which can not be calculated for non-auto-regressive models) but suffices for quantifying the effect of domain shift.

[10]Using the evaluation script from Goldberg (2019).

[11]See Appendix F for the full set of results for both **Wehbe2014** and for **Pereira2018** with similar patterns.

[12]For reflexive anaphora tasks, these trends are reversed: the models underperform the pretrained baseline and their ranking is the converse of their brain decoding scores. Reflexive Anaphora, are not explicitly annotated for in any of the three formalisms. We find, however, that they occur in a larger proportion of the sentences comprising the UCCA corpus ($1.4\%$) than those the UD ($0.67\%$) or DM ($0.64\%$) ones, indicating that domain might play a role here too.

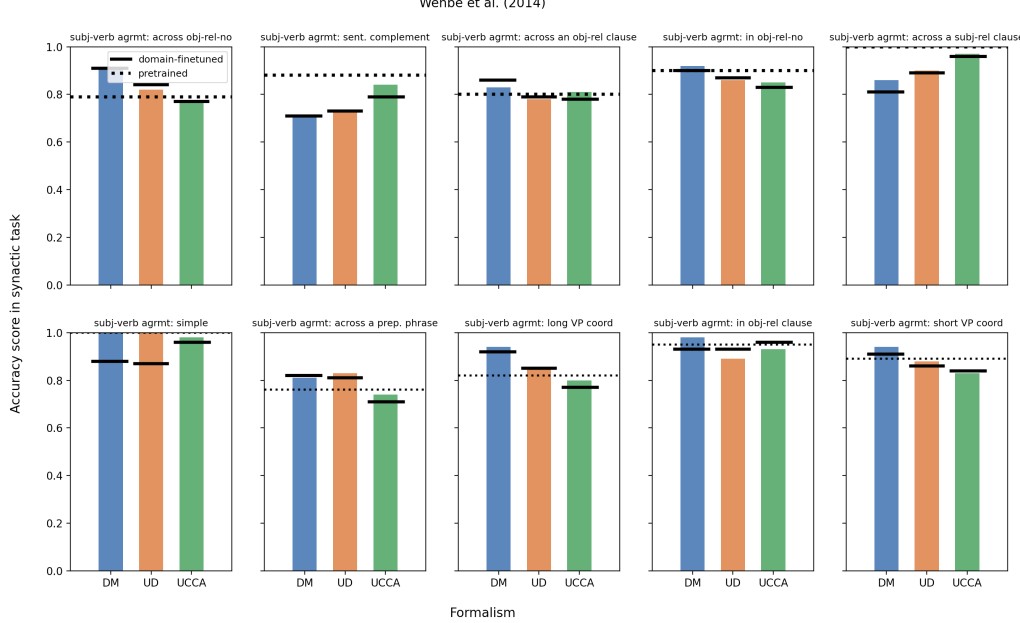

Figure 4: Accuracy per subject-verb agreement category of Marvin & Linzen (2019) for the three **Wehbe2014** models and each of the four baselines.

**Effect on semantics**    To evaluate the impact of structural bias on encoding of semantic information, we consider Semantic Tagging (Abzianidze & Bos, 2017), commonly used to analyse the semantics encoded in LM representations (Belinkov et al., 2018; Liu et al., 2019): tokens are labeled to reflect their semantic role in context. For each of the three guided attention **Wehbe2014** models and the *pretrained* model, a linear probe is trained to predict a word's semantic tag, given the contextual representation induced by the model (see Appendix E for details). For each of the three GA models, Figure 5 shows the change in test set classification F1-score,[13] relative to the *pretrained* baseline, per coarse-grained grouping of tags.[14] We find that the structural bias improves the ability to correctly recognize almost all of the semantic phenomena considered, indicating that our method for injecting linguistic structure leads to better encoding of a broad range of semantic distinctions. Furthermore, the improvements are largest for phenomena that have a special treatment in the linguistic formalisms, namely discourse markers and temporal entities. Identifying named entities is negatively impacted by GA with DM, where they are indiscriminately labeled as compounds.

**Content words and function words**    are treated differently by each of the formalisms: UD and UCCA encode all words, where function words have special labels, and DM only attaches content words. Our guided attention ignores edge labels (dependency relations), and so it considers UD and UCCA's attachment of function words just as meaningful as that of content words. Figure 8 in Appendix G shows a breakdown of brain decoding performance on content and function words for **Wehbe2014**. We find that: a) all GA models and the pretrained model show a higher function than content word decoding score, b) a large part of the decrease in decoding score of two of the three domain-finetuned baselines (UD and DM) compared to the pretrained model is due to content words.

---

[13]Note that the test set consists of 263,516 instances, therefore, the margin of change in number of instances here is considerable, e.g. $5652 * 0.6 \approx 40$ instances for the DM and UCCA models on the `temporal` category, which is the least frequent in the test set. See test set category frequencies in the appendix.

[14]The eight most frequent coarse-grained categories from an original set of ten are included—ordered by frequency from left to right; we exclude the `UNKNOWN` category because it is uninformative and the `ANAPHORIC` category because it shows no change from the baseline for all three models.

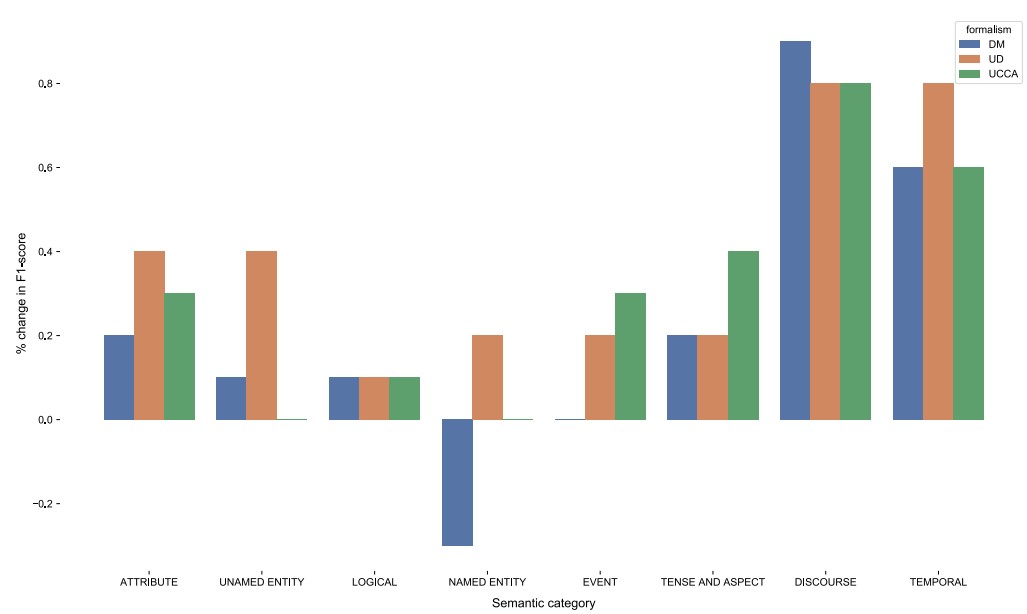

Figure 5: Change in F1-score per coarse-grained semantic class compared to the *pretrained* baseline for the three guided attention **Wehbe2014** models.

**Caveats**   The fMRI data used for both the sentence and word level analyses was recorded while participants read text without performing a specific task. Although we observe some correlates of linguistic structure, it is possible that uncovering more fine-grained patterns would necessitate brain data recorded while participants perform a targeted task. For future work it would be interesting to investigate if an analysis based on a continuous, naturalistic listening fMRI dataset (Brennan & Hale, 2019) matches up to the results we have obtained. Regarding the different linguistic formalisms, there are potential confounds such as domain, corpus size[15], and dependency length, (i.e. the distance between words attached by a relation), which depend both on the formalism and on the underlying training set text. To properly control for them, a corpus annotated for all formalisms is necessary, but such a corpus of sufficient size is not currently available.

**Conclusions**   We propose a framework to investigate the effect of incorporating specific structural biases in language models for brain decoding. We present evidence that inducing linguistic structure bias through fine-tuning using attention guided according to syntacto-semantic formalisms, can improve brain decoding performance across two fMRI datasets. For each of the 3 investigated formalisms, we observed that the models that aligned most with the brain performed best at a range of subject-verb agreement syntactic tasks, suggesting that language comprehension in the brain, as captured by fMRI recordings, and the tested syntactic tasks may rely on common linguistic structure, that was partly induced by the added attention constraints. Across formalisms, we found that models with attention guided by DM and UD consistently exhibited better alignment with the brain than UCCA for both fMRI datasets. Rather than concluding that DM and UD are more cognitively plausible, controlled experiments, with fine-tuning on each annotated corpus as plain text, suggest that the text domain is an important, previously overlooked confound. Further investigation is needed using a common annotated corpus for all formalisms to make conclusions about their relative aptness.

Overall, our proposed approach enables the evaluation of more targeted hypotheses about the composition of meaning in the brain, and opens up new opportunities for cross-pollination between computational neuroscience and linguistics. To facilitate this, we make all code and data for our experiments available at: `http://github.com/anonymized`

---

[15]It is interesting to note that decoding score rank for **Wehbe2014** corresponds to fine-tuning corpus size for the GA models (DM > UD > UCCA), but not the domain-finetuned models. A reasonable conclusion to draw from this is that dataset size might play a role in the effective learning of a structural bias.

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

## A    INJECTING STRUCTURE BY PREDICTING PARSE

One way to encode structural information from each of these formalisms into language model representations is to directly optimize for the prediction of the formalism graphs, i.e., parsing. For DM and UCCA, we use the HIT-SCIR parser Che et al. (2019), the best performing parser from the MRP 2019 Shared Task. For UD, we use the Køpsala parser Hershcovich et al. (2020) from the EUD Shared Task, which is largely based on the HIT-SCIR one. Both are transition-based parsers, which fine-tune BERT during training: BERT takes in a sequence $S$ of $P$ wordpieces and outputs a sequence of contextualized token representations $[h_1, ..., h_P]$, which the parsers use as embeddings, fine-tuning the BERT model. Our assumption is that these representations are fine-tuned during parser training to better capture the linguistic distinctions made by each formalism. After fine-tuning on each formalism's respective corpus, we extract sentence and word representations for all fine-tuned models as described above. Each of the parsers' default hyperparameters are employed.

| Model | Epoch 0 | Epoch 1 | Epoch 2 |
|---|---|---|---|
| Pereira et al. (2018) | | | |
| DM | 0.278 | 0.201 | 0.167 |
| UD | 0.277 | 0.186 | 0.159 |
| UCCA | 0.277 | 0.189 | 0.161 |
| **PRE** | 0.277 | | |
| Wehbe et al. (2014) | | | |
| DM | 0.225 | 0.126 | 0.083 |
| UD | 0.225 | 0.092 | 0.065 |
| UCCA | 0.225 | 0.110 | 0.077 |
| **PRE** | 0.226 | | |

Table 1: Brain decoding scores (Pearson's $r$) for each of the BERT models fine-tuned via **parsing**, and for the pretrained baseline (PRE). Note that the latter is not fine-tuned.

Results for the models fine-tuned via parsing show divergence in brain decoding performance. Indeed, we find that as parsing performance (as measured by unlabeled undirected attachment scores (UUAS)) improves on the held-out development set, brain decoding performance declines. This finding is congruent with the results of Gauthier & Levy (2019), which show that fine-tuning on GLUE tasks Wang et al. (2018) leads to a decline in brain decoding performance, until a ceiling point where it eventually stabilizes. In our experiments, after one epoch of fine-tuning, decoding performance is equivalent to the one achieved by the pretrained model. However, with more fine-tuning, the models consistently diverge, as shown in Table 1. These results are averaged over two fine-tuning runs. Understanding the learning dynamics that lead to such divergence is an interesting avenue for future work.

## B    MEAN/MEDIAN RANK RESULTS

Table 2 shows results for the Pearson's $r$ metric reported in the main paper, alongside the mean and median rank metrics reported in Gauthier & Levy (2019), which give the rank of a ground-truth sentence representation in the list of nearest neighbors of a predicted sentence representation, ordered by increasing cosine distance. This metric evaluates representations based on their support for contrasts between sentences/words which are relevant to the brain recordings. The table shows that the models which have higher Pearson $r$ scores, also have a lower average ground truth word/sentence nearest neighbour rank i.e. induce representations that better support contrasts between sentences/words which are relevant to the brain recordings.

| Model | Pearson's $r$ | Mean rank | Median rank |
|---|---|---|---|
| Pereira et al. (2018) | | | |
| **DF-B DM** | 0.269 | 33.58 | 12.95 |
| **DF-B UD** | 0.277 | 32.91 | 13.03 |
| **DF-B UCCA** | 0.259 | 37.05 | 15.09 |
| **GA DM** | 0.280 | 32.66 | 12.39 |
| **GA UD** | 0.286 | 30.79 | 11.44 |
| **GA UCCA** | 0.268 | 34.54 | 13.77 |
| **PRE** | 0.276 | 32.18 | 12.13 |
| Wehbe et al. (2014) | | | |
| **DF-B DM** | 0.204 | 493.11 | 89.32 |
| **DF-B UD** | 0.206 | 497.24 | 81.69 |
| **DF-B UCCA** | 0.164 | 689.89 | 227.30 |
| **GA DM** | 0.343 | 172.45 | 10.96 |
| **GA UD** | 0.280 | 255.127 | 18.28 |
| **GA UCCA** | 0.261 | 315.73 | 25.78 |
| **PRE** | 0.225 | 436.70 | 53.13 |

Table 2: Brain decoding scores as measured via three metrics — Pearson's $r$, Mean rank, and Median Rank — for each of the domain-finetuned baseline (DF-B) models, the guided attention models (GA), and the pretrained (PRE) model.

## C   SIGNIFICANCE TESTING

**Bootstrapping**   The bootstrapping procedure is described below. For each subject of $m$ subjects:

1. There are $n$ stimuli sentences, corresponding to $n$ fMRI recordings. A linear decoder is trained to map each recording to its corresponding LM-extracted (PRE, DF-B,GA) sentence representation. This is done using 12-fold cross-validation. This yields predicted a 'sentence representation' per stimuli sentence.

2. To compensate for the small size of the dataset which might lead to a noise estimate of the linear decoder's performance, we now randomly resample $n$ datapoints (with replacement) from the full $n$ datapoints.

3. For each resampling, our evaluation metrics (pearson's $r$, mean rank, etc.) are computed between the sampled predictions and their corresponding 'gold representations', for all sets of LM reps. We store the mean metric value (e.g. pearson r score) across the $n$ 'sampled' datapoints. We run 5000 such iterations.

4. This gives us 5000 such paired mean (across the $n$ samples, that is) scores for all models.

5. When comparing two models, e.g. **GA DM** vs.**PRE**, to test our results for strength of evidence of generalization over stimuli, we compute the proportion of these 5000 paired samples where e.g. **GA DM**'s mean sample score is greater than **PRE**. After Bonferroni correction for multiple hypothesis testing, is the $p$-value we report. See 3 for these per subject $p$-values for **Pereira 2018**. For **Wehbe 2014**, comparisons between each of the GA models and the pretrained baseline lead to $p = 0.000$ (i.e. The GA model mean score is greater than the pretrained baseline's mean score for all 5000 sets of paired samples), for all subjects. We, therefore, do not include a similar table.

6. We average over these 5000 samples per subject, and use these $m$ subject means for the across-subject significance testing, which is described below.

**Strength of generalization across subjects**   To test our results for strength of generalization across subjects, we apply the Wilcoxon signed rank test (Wilcoxon, 1992) to the $m$ by-subject mean scores (see above), comparing the GA models to the pretrained baselines. Since $m = 8$ for both datasets, the lowest $p$-value is 0.0078 (if every subject's difference score consistently favors the GA model over the baseline or vice versa).

In the case of **Pereira 2018**: for **PRE** vs. **GA UD** we get a $p$-value of 0.0078 (0.0234 after Bonferroni correction); for **PRE** vs. **GA DM** we get an $p$-value of 0.015 (0.045 after Bonferroni correction); for

**PRE** vs. **GA UCCA** we get a $p$-value of 0.0078 (0.0234 after Bonferroni correction, here **PRE** > **GA UCCA** for all subjects).

In the case of **Wehbe 2014**: all comparisons yield a $p$-value of 0.0078 (0.045 after Bonferroni correction), where the $GA$ model > the pretrained baseline.

| | | | | Pereira et al. (2018) | | | | |
|---|---|---|---|---|---|---|---|---|
| Model/Subject | M02 | M04 | M07 | M08 | M09 | M14 | M15 | P01 |
| **GA UD** | 0.000 | 0.110 | 0.011 | 0.021 | 0.000 | 0.009 | 0.000 | 0.039 |
| **GA DM** | 0.132 | 0.216 | 0.031 | 0.014 | 0.000 | 0.417 | 0.186 | 0.085 |
| **GA UCCA** | 0.014 | 0.015 | 0.052 | 0.041 | 0.452 | 0.002 | 0.000 | 0.003 |

Table 3: $p$-values resulting from paired bootstrap test described above, for each of the three GA models when compared to the pretrained baseline.

## D  THE DOMAIN EFFECT

Table 4 shows average word perplexity scores for the pretrained model and the domain-finetuned models for each of the three text domains on the stimuli from **Pereira2018** and **Wehbe2014**. Scores are averaged over the words in a sentence and the sentences (stimuli) in the datasets.

| Pereira et al. (2018) | |
|---|---|
| **PRE** | 14.09 |
| **DF-B DM** | 19.11 |
| **DF-B UD** | 19.08 |
| **DF-B UCCA** | 20.67 |
| **GA DM** | 20.82 |
| **GA UD** | 17.15 |
| **GA UCCA** | 17.47 |
| Wehbe et al. (2014) | |
| **PRE** | 34.79 |
| **DF-B DM** | 36.11 |
| **DF-B UD** | 38.41 |
| **DF-B UCCA** | 40.45 |
| **GA DM** | 33.24 |
| **GA UD** | 37.16 |
| **GA UCCA** | 33.60 |

Table 4: Average word perplexity scores for each of the domain-finetuned baseline (DF-B) models, the guided attention models (GA), and the pretrained (PRE) model.

## E  SEMANTIC TAGGING

**Probing details**   Representations for the probing task are derived as described in 3.1 for each sentence in the development and testing sets from Abzianidze & Bos (2017). The development set is employed as a training set, because it is mostly manually annotated/corrected (as opposed to the much noisier training set) and because it is already possible to train rather accurate semantic taggers which suffice for our analysis with a training set of that size (131337 instances). We report results for the official test set. Table 5 shows the frequency of each semantic category we report scores for in the test set. An $L2$ regularised logistic regression model is utilised.

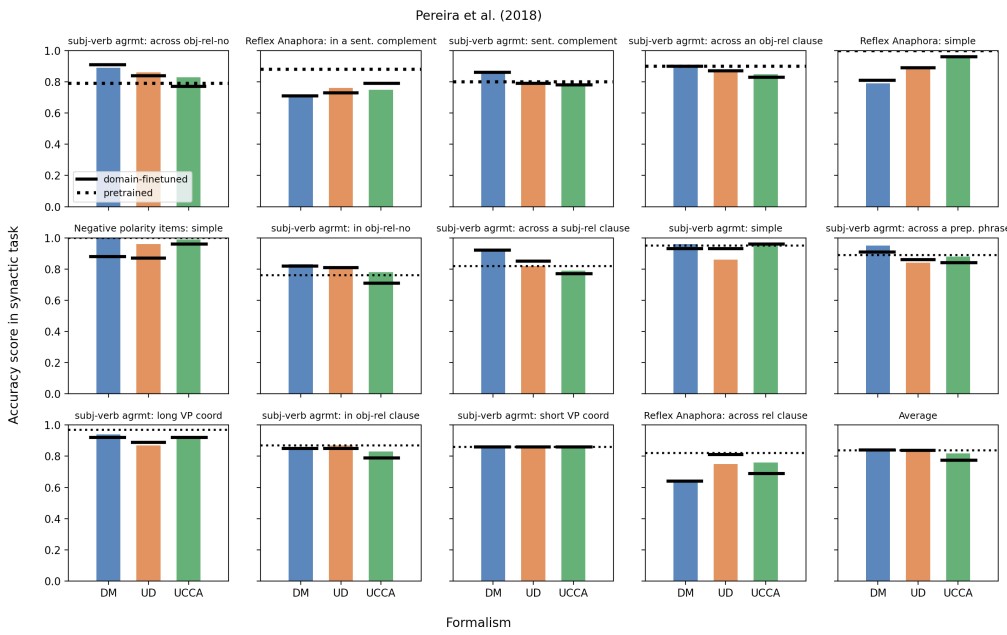

Figure 6: Targeted syntactic evaluation accuracy scores per category for **Pereira2018** models.

**Further discussion** We observe the largest improvements for the `DISCOURSE` and `TEMPORAL` categories. The former involves identifying subordinate, coordinate, appositional, and contrast relations. These relations are highly influenced by context, and correctly classifying them can often be contingent on longer dependencies, which the structural bias increases 'awareness' of. The `TEMPORAL` category, on the other hand, consists of tags such as `clocktime` or `time of day` which are applied to multi-word expressions, e.g *27th December*. Highlighting these dependencies by assigning more weight to the attention between their sub-parts is likely helpful for their accurate identification.

| Category / Frequency | |
|---|---|
| Attribute | 63763 |
| Unamed Entity | 48654 |
| Logical | 32973 |
| Named Entity | 29271 |
| Event | 25338 |
| Tense and Aspect | 15208 |
| Discourse | 9948 |
| Temporal | 5652 |

Table 5: Semantic category frequency in the test set.

## F   TARGETED SYNTACTIC EVALUATION SCORES

Figures 6 and 7 show the performance of the **Pereira2018** and **Wehbe2014** models and the four baselines for each of the syntactic categories from Marvin & Linzen (2019).

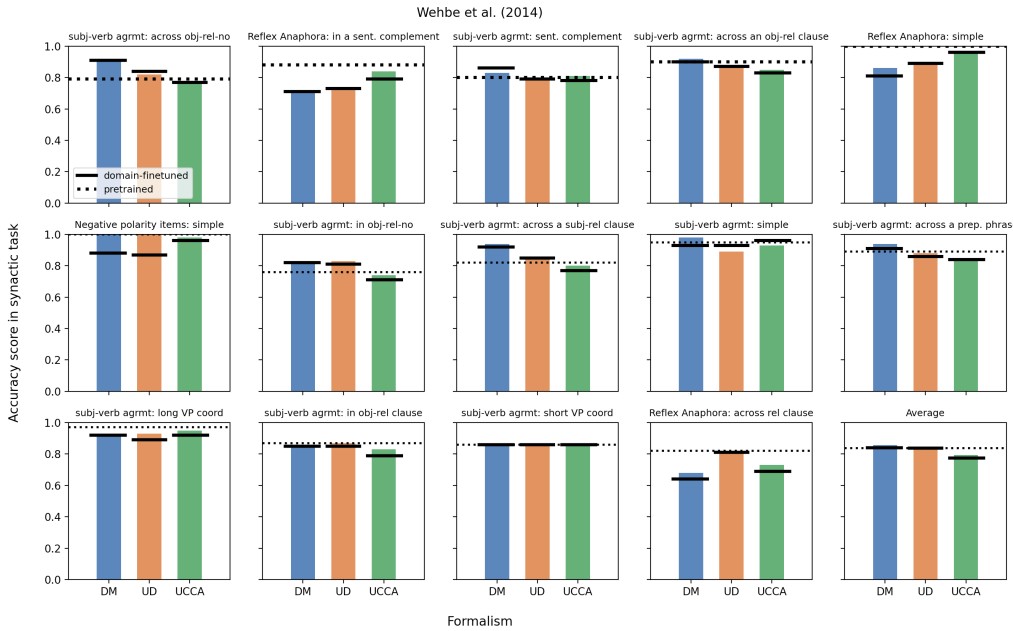

Figure 7: Targeted syntactic evaluation accuracy scores per category for **Wehbe2014** models.

## G CONTENT WORDS AND FUNCTION WORDS ANALYSIS

Figure 8 shows the breakdown of brain decoding accuracy by content and function words for **Wehbe2014**. We consider content words as words whose universal part-of-speech according to spaCy is one of the following: {ADJ, ADV, NOUN, PROPN, VERB, X, NUM}. Out of a total of 4369, 2804 are considered content words and 1835 as function words.

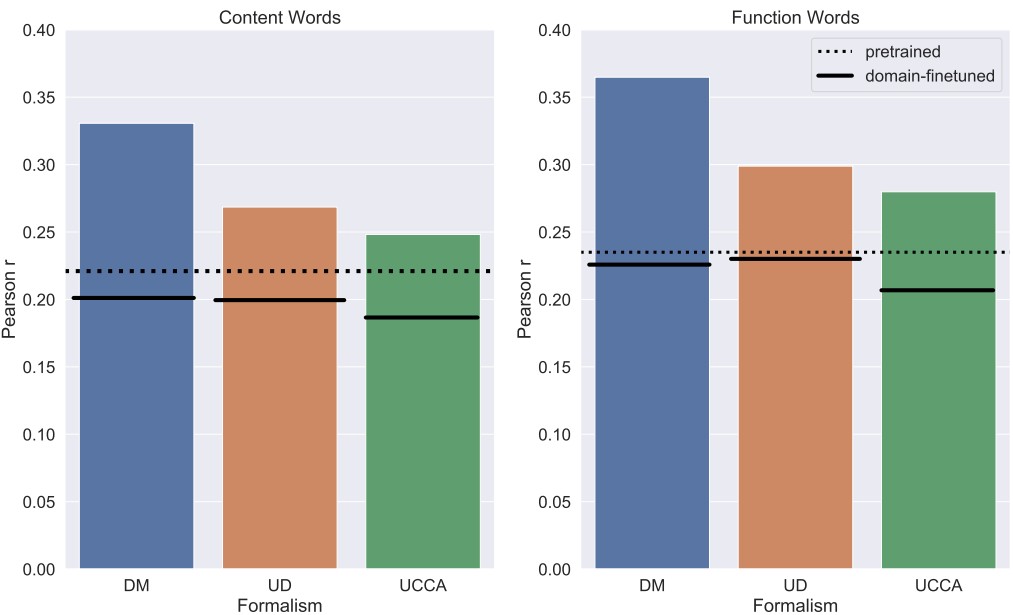

Figure 8: Content word and function word brain decoding score (mean Pearson's $r$) for all models fine-tuned by MLM with guided attention on each of the formalisms (points), as well the four baselines: pretrained BERT, dotted line), and the domain-finetuned BERT by MLM on each formalism's training text without guided attention (solid lines).

