# OpenReview forum: "Does injecting linguistic structure into language models lead to better alignment with brain recordings?"
_ICLR.cc/2021/Conference — Reject_

### Official Review · AnonReviewer3 · 2020-10-28
**Provocative paper, but several technical concerns**

**Rating:** 6
**Confidence:** 3

**Review:**

Summary of paper: the authors explore adding a soft structural attention constraint to BERT, by penalizing attention weights that are substantially different from a head–dependent "adjacency" matrix derived from dependency parses.  BERT is then fine-tuned with and without ("domain-finetuned") this constraint on corpus data for which fMRI recordings from participants during reading are available. A linear classifier from the final layer of BERT's embedding (mean-pooled) is then learned to the fMRI data.  Within this pipeline, domain-finetuned models are not an improvement over unfinetuned BERT, but fine-tuning with the structural attention constraint improves decoding to fMRI data, especially for word-level data (the Wehbe2014 dataset).

Assessment: this is a nice paper that investigates an intuitive method of incorporating syntax-based, structural soft attention constraints into Transformer encoder models for language.  What makes the contribution fairly distinctive is evaluation on alignment with human fMRI recordings during comprehension of the texts.  The results show improvements in decoding relative to baseline models that involve no fine-tuning and/or domain-adaptation fine-tuning alone (no structural attention constraints), especially for fMRI data that are recorded below the sentence level.  The authors also evaluate the effect of fine-tuning on targeted syntactic evaluations from Marvin & Linzen; the results here are not particularly conclusive.  Overall, this is a potentially solid, if not ground-breaking, contribution.  However, there are a number of technical questions that are left unclear in the submission, and some of the results are cause for some concern.  These concerns need to be addresed in order for the submission to be fully satisfactory.

The single biggest concern is the extraordinarily high word perplexity scores in Table 2 for Wehbe2014 -- which get much, much worse after fine-tuning.  It is important to understand what's going on here in order to make sense of the core potential contribution of the paper, because it's only in the Wehbe2014 dataset where there seem to be appreciable improvements in decoding performance.  I would guess that the high perplexity comes from poor prediction of the proper nouns in the Harry Potter book chapter.  Maybe there needs to be some amount of fine-tuning of the models to the domain of the test-set corpus.  Overall, the paper needs more clarity on why it is only the Wehbe2014 dataset where the perplexity is so high and the fine tuning affects decoding performance so much.

Additional technical questions:

1) How is the split of a word into word pieces handled in the adjacency matrix representing word–word dependencies?

2) How are the adjacency matrix and each head's attention weight matrix converted into a distribution for computing cross-entropy loss?  Are the entries normalized globally? By row? By column?

3) What are the perplexities like for domain-finetuned (no structural attention constraint) BERT?  These are missing from Table 2 (Appendix B), but are potentially important in interpreting your results.

4) What words are pooled over for the Wehbe2014 analyses -- the four words in the 2-second window?

5) Section 4.1 reports that UD and DM finetuned models are significantly better in brain decoding than the un-finetuned baseline, at p<0.0001, but the 95% confidence intervals for subject scores look very different.  And the difference in mean decoding performance for DM finetuning is barely visible.  How are you computing your confidence intervals and your p-values?  Why are they so different, and how are you getting such high confidence in improvements over the unfinetuned baseline here?

6) How do your results compare to those using the best fine-tuning methods from Gauthier & Levy (2019), which involve scrambling the input sentences?

7) Given that in Wehbe2014 each fMRI image corresponds to four words, most of which probably contain both function and content words, how is the content/function word analysis defined and performed?

Additional comments:

* the authors write that "increase in perplexity roughly corresponds to lower brain decoding scores", but this doesn't look consistent with Table 2 and Figure 3.  For Wehbe2014, UCCA data yield the worst decoding accuracy but yield better perplexity than DM data, which yield decoding accuracy only slightly worse than the UD data.  The monotonicity is cleaner for Pereira2018 but the overall differences in decoding performance are much smaller.

---

> ### Author Response · Authors · 2020-11-15
> **Response to reviewer #3: thanks, clarifications (Part 1)**
>
> Dear reviewer #3, we truly appreciate your comprehensive and thoughtful review, which has already helped us improve this work. Please let us know if you have any additional comments or questions.
>
> #### Regarding word perplexity, there are two important clarifications to make: ####
>
> - We have found an explanation for the anomalously high perplexity scores. In the results reported in Table 2, the exponentiation of the log-likelihood term is being applied per sentence (i.e. over the average word log-likelihood per sentence), rather than over the entire dataset.
>
> - The results in Table 2 are actually for the domain-finetuned baselines, i.e. the models fine-tuned on each formalism’s corpus without the structural attention constraint. This was not sufficiently clear.
>
> We have now adjusted the method by which the perplexity was being calculated, and included the results for both the domain-fintuned baselines and the structurally biased models. Please find the results below (we will also update them in the appendix):
> * PRE.: pretrained
> * DF-B: domain-finetuned baseline
> * GA: guided attention finetuning
>
>
> *Pereira et al. (2018)*
> -----------------------     -------
> **PRE**                  14.09
> -----------------------    -------
> **DF-B DM**         19.11
>
> **DF-B UD**      19.08
>
> **DF-B UCCA**     20.67
> -----------------------    -------
> **GA DM**        20.82
>
> **GA UD**          17.15
>
> **GA UCCA**    17.47
>
> *Wehbe et al. (2014)*
> ----------------------- -------
> **PRE**          34.79
> ----------------------- -------
> **DF-B DM**        36.11
>
> **DF-B UD**         38.41
>
> **DF-B UCCA**    40.45
> ----------------------- -------
> **GA DM**         33.24
>
> **GA UD**        37.16
>
> **GA UCCA**    33.60
>
> We now observe the following:
>
> - Our main conclusion re. the effect of domain remains unchanged: simply running MLM finetuning on each of the texts of the three datasets (UD, DM, and UCCA) leads to higher perplexity scores on the fMRI stimuli texts. Moreover, except in the case of DM for Pereira 2018, the models finetuned via  MLM + guided attention (GA), have lower perplexities than their domain-finetuned baseline counterparts.
>
> - As you correctly note, there is, overall, no clear correspondence between lower perplexity and higher brain decoding scores -- although we find a tendency for the domain-finetuned baselines, where a higher decoding score (descending rank, P2018: UD > DM > UCCA; W2014: DM > UD > UCCA) corresponds to a lower perplexity (ascending ranking, P2018: UD > DM > UCCA; W2014: DM > UD > UCCA). This does not hold for the structurally biased models (as domain is, perhaps, no longer the primary factor involved).

---

> > ### Author Response · Authors · 2020-11-15
> > **Response to reviewer #3: Answers to technical questions (Part 2)**
> >
> > ##### Answers to technical questions: #####
> >
> >
> > *How is the split of a word into word pieces handled in the adjacency matrix representing word–word dependencies?*
> > - We align word pieces with their corresponding words, then each word piece that is part of a word is included in the dependency. e.g. if Word1 is made up of {w-pieceA, w-pieceB} and Word2 of {w-pieceF, w-pieceG} and Word1 and Word2 have a dependency in the word adjacency matrix, then the word piece adjacency matrix we build will have w-pieceA and w-pieceB each connected to both w-pieceF, and w-pieceG (and vice-versa, since edge directionality is not considered in our setup).
> >
> > *How are the adjacency matrix and each head's attention weight matrix converted into a distribution for computing cross-entropy loss? Are the entries normalized globally? By row? By column?*
> >  - The entries are normalized globally, i.e. across the entire attention weight matrix.
> >
> > *What are the perplexities like for domain-finetuned (no structural attention constraint) BERT? These are missing from Table 2 (Appendix B), but are potentially important in interpreting your results.*
> > - Please see above (part 1 of response).
> >
> > *Section 4.1 reports that UD and DM finetuned models are significantly better in brain decoding than the un-finetuned baseline, at p<0.0001, but the 95% confidence intervals for subject scores look very different. And the difference in mean decoding performance for DM finetuning is barely visible. How are you computing your confidence intervals and your p-values? Why are they so different, and how are you getting such high confidence in improvements over the unfinetuned baseline here?*
> > - The p-values are calculated on the basis of a Wilcoxon signed rank test, applied to the results of 3000 iterations of bootstrap resampling of the mean (across cross-validation splits) pearson’s r score per subject. So the differences between the DM/UD models and the pretrained baseline are significant at  p<0.0001 per subject. The plot shows the mean decoding scores across cross-validation splits, bootstrap iterations, linear decoder runs, MLM (+ guided-attention) finetuning runs, and finally subjects. The confidence intervals are calculated based on this final quantity (subject scores), which is not the same quantity as the quantity used for computing the p-values we report (which are per subject). We will clarify this.
> >
> > *How do your results compare to those using the best fine-tuning methods from Gauthier & Levy (2019), which involve scrambling the input sentences?*
> > - In initial experiments, we find that using final layer hidden state mean pooling instead of the ‘[CLS]’ token yields representations which can be mapped with significantly higher Pearson’s r and lower Average Rank (AR, please see response to reviewer four for description) scores on Pereira 2018. Therefore, the decoding scores we report for the baseline ‘pretrained BERT’ are already an ‘improvement’ compared to the best finetuning results reported in Gauthier & Levy (2019), which we also independently replicate (pearson’s r ≈ 25.5 vs. pearson’s r ≈ 27.6  || AR ≈ 32 vs. ≈ 38)).  We do not subsequently run a direct comparison of finetuning methods -- that is, using an equivalent representation extraction method -- because the conclusions which can be drawn from such a comparison are not immediately transparent: changes in decoding score, would likely be due to the influence of a variety of different factors in the two experiments. There is, however, potentially a rather interesting connection to be explored there, since there is recent work ([1]) showing that ‘composition’ drives the brain’s language network, even with the violation of grammatical word order restrictions, as long as local dependencies are preserved.
> >
> > *Given that in Wehbe2014 each fMRI image corresponds to four words, most of which probably contain both function and content words, how is the content/function word analysis defined and performed?*
> > - During decoding, each ‘target representation’ (BERT hidden state corresponding to a word) either represents a content word or a function word. Although the source representation (fMRI recording corresponding to a two second time interval and four words) does more likely than not include a combination of both content and function words, the analysis is conducted on the basis of the target representations, and therefore it is possible to separately evaluate the decoding performance for each word contained in a given time interval, e.g. how well can the hidden representation of each word in the phrase ‘win that Quidditch Cup’ be decoded from the fMRI recording representing the phrase.
> >
> > 1: https://www.mitpressjournals.org/doi/full/10.1162/nol_a_00005

---

> > > ### Comment · AnonReviewer3 · 2020-11-15
> > > **A couple of follow-up questions**
> > >
> > > Thank you for the thoughtful response! I have a follow-up question regarding your p-values:
> > >
> > > >>Section 4.1 reports that UD and DM finetuned models are significantly better in brain decoding than the un-finetuned baseline, at p<0.0001, but the 95% confidence intervals for subject scores look very different. And the difference in mean decoding performance for DM finetuning is barely visible. How are you computing your confidence intervals and your p-values? Why are they so different, and how are you getting such high confidence in improvements over the unfinetuned baseline here?
> > >
> > > >The p-values are calculated on the basis of a Wilcoxon signed rank test, applied to the results of 3000 iterations of bootstrap resampling of the mean (across cross-validation splits) pearson’s r score per subject. So the differences between the DM/UD models and the pretrained baseline are significant at p<0.0001 per subject. The plot shows the mean decoding scores across cross-validation splits, bootstrap iterations, linear decoder runs, MLM (+ guided-attention) finetuning runs, and finally subjects. The confidence intervals are calculated based on this final quantity (subject scores), which is not the same quantity as the quantity used for computing the p-values we report (which are per subject). We will clarify this.
> > >
> > > I am still confused. Let's focus on Pereira2018 since that is the dataset where the decoding performance changes seems so small. This dataset has 8 subjects and I understand you are doing 12-fold cross-validation (for each subject, is that right?). What exactly is getting resampled in the bootstrapping process? What is the Wilcoxon signed rank test being applied to? And what does "p<0.0001 per subject" mean? How does your procedure test for strength of generalization across subjects?
> > >
> > > To approach this from a different angle -- a simple test you could do is a paired t-test between the by-subject mean performance of (i) the GA model, and (ii) the model you're comparing it with. So, since you have 8 subjects worth of data you would have 8 pairs of values. Or you could use the signed rank test on these 8 subjects worth of data. From my own quick check, the lowest p-value you could get with the Wilcoxon signed rank test for 8 subjects is p=0.008 (if every subject's difference score consistently favors the GA model over the baseline) so I'm concerned that you may not be adequately testing for strength of generalization across subjects.

---

> > > > ### Author Response · Authors · 2020-11-17
> > > > **Response to follow-up questions**
> > > >
> > > > Thank you for following up!
> > > >
> > > > You are correct in thinking that our statistical testing does not directly address generalization across subjects. Generalization across subjects is notoriously difficult in brain imaging studies, due to small sample size, anatomical differences between subjects, and the fact that neural response patterns can be highly diverse across subjects. The statistical tests we report aim to test for significance per subject. The exact procedure we carry out (described specifically for Pereira 2018) is as follows:
> > > >
> > > > For each subject:
> > > > 1. There are 384 stimuli sentences, corresponding to 384 fMRI recordings, a linear decoder is trained to map each recording to its corresponding LM-extracted (PRE, DF-*,GA-*) sentence representation. This is done using 12-fold cross-validation. This yields predicted ‘sentence representation’ per stimuli sentence.
> > > > 2. To compensate for the small size of the dataset which might lead to a noise estimate of the linear decoder’s performance, we now randomly resample 384 datapoints (with replacement) from the full 384 datapoints.
> > > > 3. For each resampling, our evaluation metrics (pearson r, average rank, etc.) are computed between the sampled predictions and their corresponding ‘gold representations’, for all sets of LM reps. We store the mean metric value (e.g. pearson r score) across the 384 ‘sampled’ datapoints. We run 3000 such iterations.
> > > > 4. This gives us 3000 such paired scores across models.
> > > > 5. We now run the signed rank test to test whether a given two models’ scores were drawn from populations having the same distribution. This returns a p-value.
> > > > 6. After applying the Bonferroni correction for multiple hypothesis testing, this is the p-value we report.
> > > >
> > > > Having said that, if we run the analysis you describe, applying a signed rank test to the by-subject mean scores below:
> > > >
> > > > | model/subject 	| M2    	| M4    	| M7    	| M8    	| M9    	| M14   	| M15   	| P01   	|
> > > > |---------------	|-------	|-------	|-------	|-------	|-------	|-------	|-------	|-------	|
> > > > | **PRE**       	| 0.312 	| 0.258 	| 0.285 	| 0.266 	| 0.246 	| 0.217 	| 0.285 	| 0.342 	|
> > > > | **GA UD**     	| 0.325 	| 0.267 	| 0.294 	| 0.274 	| 0.259 	| 0.230 	| 0.286 	| 0.350 	|
> > > > | **GA DM**     	| 0.316 	| 0.263 	| 0.292 	| 0.269 	| 0.253 	| 0.223 	| 0.282 	| 0.345 	|
> > > > | **GA UCCA**  	| 0.303 	| 0.247 	| 0.279 	| 0.262 	| 0.240 	| 0.211 	| 0.271 	| 0.334 	|
> > > >
> > > >
> > > > - For PRE vs. **GA UD**  we get an uncorrected p-value of 0.0078 (in line with your calculation, **GA UD** > PRE for all subjects)
> > > > - For PRE vs. **GA DM** we get an uncorrected p-value of 0.015
> > > > - For PRE vs. **GA UCCA** we get an uncorrected p-value of  0.0078 (here PRE > **GA UCCA** for all subjects)

---

> > > > > ### Comment · AnonReviewer3 · 2020-11-17
> > > > > **Response re: statistical testing**
> > > > >
> > > > > Thank you, this is all very helpful! I think I understand better. If you are running the signed rank test on the set of 3000 paired scores resulting from bootstrapping, I think that is an incorrect approach. As an example, suppose you only had three stimulus sentences, and the GA UD - PRE rank difference scores for these three stimulus sentences are 0.05, 0.04, -0.02 (modulo some multiplicative constant depending on how you're representing rank scores). In all likelihood, over 80% of your samples would have a mean difference score above 0, and the signed rank test would come out highly significant. Here is a bit of simple R code showing this:
> > > > >
> > > > >     item_difference_scores <- c(0.05,0.04,-0.02)
> > > > >     wilcox.test(item_difference_scores) # clearly as far from significance as possible
> > > > >     samples <- sapply(1:3000,function(ignore) mean(sample(xx,3,replace=TRUE)))
> > > > >     mean(samples>0)
> > > > >     wilcox.test(samples) # comes out highly significant
> > > > >
> > > > > This approach is **not valid**: it is effectively treating the 3000 samples as iid, whereas they are definitely not.
> > > > >
> > > > > Normally one would use the bootstrap to get to a p-value by generating a bootstrap distribution of the test statistic of interest under the null hypothesis that neither of the two models is appreciably better than the other. If you are getting a statistically significant result under a nonparametric test like signed rank, the bootstrap is probably not necessary.
> > > > >
> > > > > 384 items is not such a small number of items. What results do you get when you apply the signed rank test directly to the set of 384 difference scores for each model pair?
> > > > >
> > > > > I do think you need to include by-subject analyses along the lines of what you have above. The big picture here is that you need to test your results for strength of evidence of generalization over both subjects and stimulus sentences.

---

> > > > > > ### Author Response · Authors · 2020-11-17
> > > > > > **Another response re: statistical testing**
> > > > > >
> > > > > > We apologize for the confusion arising from our compounded test - thanks a lot for spotting this!
> > > > > > Below, we report corrected p-values from the signed rank test applied directly to the 384 scores for for the **GA UD**vs.**PRE** and the **GA UD**vs.**PRE** comparisons, per subject:
> > > > > >
> > > > > > | model/subject        	| M2      	| M4    	| M7    	| M8    	| M9       	| M14     	| M15   	| P01   	|
> > > > > > |----------------------	|---------	|-------	|-------	|-------	|----------	|---------	|-------	|-------	|
> > > > > > | **GA UD** vs.**PRE** 	| 5.62e-5 	| 0.331 	| 0.011 	| 0.045 	| 1.41e-08 	| 0.0076  	| 0.85  	| 0.065 	|
> > > > > > | **GA DM** vs.**PRE** 	| 0.25    	| 0.25  	| 0.076 	| 0.038 	| 0.0001   	| 0.849   	| 0.56  	| 0.11  	|
> > > > > >
> > > > > > While we find that while **GA UD** is still significantly better for most subjects, the same is not true for **GA DM** (**GA UCCA**, meanwhile, is unsurprisingly, significantly worse for most subjects).  For the sake of completeness, we apply this same procedure to Wehbe 2014, getting corrected p-values that are << 0.001 for all subjects across the three **GA** models .
> > > > > > We will update both the p-values and procedure reported in the paper. We will also include the results for generalisation across subjects.

---

### Official Review · AnonReviewer2 · 2020-10-28
**An interesting paper with an interdisciplinary appeal. In general well-executed, well-written study, with some (minor) issues.**

**Rating:** 7
**Confidence:** 4

**Review:**

This paper tests whether fine-tuning large pre-trained language models
(LMs) with structural information can increase the correlation between
these representations and the representations of brain activity
measured while processing the same stimuli. The injection of the
structural information is done through fine-tuning of the pre-trained
model by "guided attention", which makes use of binary relations
between the words according to three different syntactic or semantic
formalisms. The authors map the brain activity to each of the
alternative LM representations via a regression model, and measure the
alignment by using correlation between the predicted (from brain
activity) and actual output of the alternative models. The results
show that under certain conditions representations learned through
guided attention aligns better with the representations of brain
activity.

In general the investigates an interesting question which may be
(eventually) relevant to both understanding the way humans process
language, and possibly building better computational models. The
method followed in the study is (mostly) sound, and the paper is
written well.

A potential issue with the method is the direction of the prediction
in "brain decoding" regression (section 3.5). Authors predict the
model representations from the "brain representation" (this seems to
be based on earlier studies, but I did not verify). In my opinion the
reverse is more meaningful. Since the invariant quantity in this study
is the representations coming from the brain imaging. This is
important, because the success of the regression is not only about the
amount of information in the predictors, but also simplicity of the
task. Hence, an alteration of the model representations that
simplifies them may result in better predictions, and hence, higher
correlations

Except the above, I have some (mostly minor) comments:

- I would be happier with a bit more explicit discussion of the main
  results. After reading the articles, I am still not sure what to
  take away from the main experiments. The effect on two different
  data sets (also means representations at different levels/units) are
  quite different - not allowing a clear conclusion. Side issues
  discussed (the effects of the use of different formalisms, the
  effect of domain, particular syntactic patterns, content vs.
  function words are also relatively brief and far from being
  conclusive). I think a clearer discussion of the main results, and
  investigation of reasons for the discrepancy between the data sets
  would make the paper stronger.

- It would help if the data is explained slightly better.
  Particularly, it would make the paper more self contained if the
  authors specify whether any of the data sets (section 3.3) had
  automatic annotation. On a somewhat related note, comparisons
  between the formalisms seem to correlate with the data sizes, which
  is not pointed out in the paper.

- A few language/typography issues/suggestions:

    - I am not sure about the ICLR guidelines, but avoiding citations
      in the abstract is a good idea (abstracts should stand alone).
    - Footnote marks should go after punctuation (footnote mark 8)
    - Conclusions line 3: "attention guided" -> "guided attention" ?
    - There are case (normalization) issues in the references:
      "groningen", "erp", "bert" (not exhaustive, a through check is
      recommended).

---

> ### Author Response · Authors · 2020-11-18
> **Response to reviewer #2**
>
> Dear reviewer #2, we thank you for your appreciation of our work and your helpful comments/suggestions, which we aim to address:
>
> - Regarding the direction of prediction in the regression between the brain and model representations: in addition to comparing the regression performance during brain decoding, we have also evaluated all models on a range of syntactic probing tasks proposed by Marvin & Linzen (2019). From these evaluations, we observe that after attention-guided fine-tuning: a) two of the guided-attention models have a higher score than the pretrained baseline and the domain-finetuned baselines for most tasks and b) the ranking of the models corresponds to their ranking on the brain decoding task (DM > UD > UCCA). Taking both the brain decoding results and these syntactic probing results together, we argue that the guided-attention has altered the model representations in a beneficial way that is beyond just simplifying the representations in a task-irrelevant way. However, we agree with the reviewer that investigating the opposite direction of prediction (from the model representations to the brain representations) is also informative, and indeed this is a recently popular direction (Toneva and Wehbe, 2019; Schwartz et al. 2019, Schrimpf et al. 2020) that will make for excellent future application of our proposed method.
> - We will clarify and consolidate our discussion sections to better highlight the conclusions.
> - Regarding the data, it is all manually annotated by expert annotators.. We had cut the section short to save space, but will now include the additional information. Fine-tuning data size is indeed correspondent to decoding score (for Wehbe 2014) and even to performance on (most of) the subject-verb agreement tasks. We will add mention of this.

---

### Official Review · AnonReviewer1 · 2020-10-29
**Does it work the same way on other LM?**

**Rating:** 7
**Confidence:** 3

**Review:**

An interesting paper that discusses whether injecting three types of syntactic and semantic formalisms lead to better alignment with how language is processed in the brain. The authors conduct experiments with the BERT model and two fMRI datasets and show that including linguistic structure through fine-tuning can improve brain decoding performance.

The paper would be improved by experimenting with language models other than BERT, as it is not clear at the moment whether the produced results are generalizable to different language models or are BERT-specific. For example, additional experiments with AlBert, distilBert and RoBerta would provide additional insights on the effect of size of the model, in terms of the number of parameters. Comparison of Bert to GPT and XLNet would emphasize the advantages/disadvantages of autoencoder-based vs autoregressive models and could potentially provide additional insight on how attention is represented in human brain.

It would also be interesting to read a discussion of semantic analysis, as currently the paper concentrates the most on syntactic formalism as represented in both BERT and fMRI data. Specifically, it would be interesting to know if the injection of syntax impacts the semantic representations. One of the possible methods to measure that would be probing for semantic classes (as in Yaghoobzadeh et al., 2019. Probing for Semantic Classes)

---

> ### Author Response · Authors · 2020-11-16
> **Response to Reviewer #1**
>
> Dear Reviewer #1, we thank you for your appreciation of our work and for your helpful suggestions on how to improve it.
>
> We agree that expanding the scope of the experiments to other language models could potentially yield interesting conclusions regarding the interaction of structural bias with model size and architecture. We made a conscious choice to focus in this work on evaluating across multiple linguistic formalisms and on presenting results for more than one imaging dataset, since these two facets of our investigation were more immediately crucial to the core research questions. However, we see extensions along the ‘architecture/training objective’ dimension as an important next step that we would very much like to address in follow-up work.
>
> Analysing the effect of structural bias on the models’ encoding of semantics could indeed potentially allow for a deeper understanding of the factors which lead to a better alignment with the brain recording data. The task proposed in Yaghoobzadeh et al., 2019 appears to perhaps be better suited for non-contextualized word embeddings, than for contextualized ones. However, we are currently running an analysis using the Semantic Tagging task ([1]), which involves assigning a one of 80 fine-grained ‘semantic tags’  which cover a broad range of semantic classes (e.g.: discourse relations, logical semantics Anaphora, named entities, etc.), and describe the “semantic contribution of the token with respect to the meaning of the source expression”. We will report the results of this analysis and include it in the paper over the next few days. We thank you for the suggestion!
>
> 1: https://www.aclweb.org/anthology/W17-6901/

---

### Official Review · AnonReviewer4 · 2020-10-29
**Review of "DOES INJECTING LINGUISTIC STRUCTURE INTO LANGUAGE MODELS LEAD TO BETTER ALIGNMENT WITH BRAIN RECORDINGS?"**

**Rating:** 5
**Confidence:** 3

**Review:**

This paper describes experiments that inject linguistic information (for example dependency structures) into BERT, then measure improvements in correlation with FMRI measurements of humans reading an underlying sentence (which is also analyzed by BERT). Linguistic information is incorporated by biasing attention heads to line up with dependency (or other) structures.

Positives about the paper: it's an interesting experiment to try, and an important direction of work.

Negatives:

* It's a somewhat small increment over previous work, not sure it merits a full conference paper. As it stands the paper presents the approach and results, with little inspection of why improvements are seen. I would like the authors to go much deeper with the analysis. Are there particular syntactic constructions that are being better modeled? Is the new model much more sensitive to long range dependencies, as found in syntactic structures? Are particular classes of words effected more than others? Answering these questions will be challenging but would add a lot to the paper.

* Most importantly, the evaluation metrics are unclear. The critical sentence in the paper is "To evaluate the regression models, Pearson’s correlation coefficient between the predicted and the corresponding heldout true sentence or word representations is computed". This is a terse description of a critical part of the approach, and I can't make sense of it.

Part of my unease about the evaluation is the following. The matrix $D_{fr}$ is the output from BERT. Importantly in The definition of L_{ifr} this matrix is predicted from the "brain" matrix B_i. If $D_{fr}$ was all zeros it would be trivially predictable (and hence gameable). In the original Gauthier and Levy paper they appear to use metrics in addition to MSE. In this paper some variant of Pearson's correlation coefficient is used - but I can't understand what exactly this is, and my worry is that it is trivially gameable in the same way as MSE.

---

> ### Author Response · Authors · 2020-11-12
> **Response to reviewer 4: thanks and comments**
>
> Dear Reviewer #4, we thank you for your helpful comments and feedback.
>
> Regarding the evaluation metrics:
>
> We report pearson’s r correlation, employing it as a bounded, invariant measure of representational similarity. In general, this is of course, yes, vulnerable to ‘trivial gaming’, as instanced in your all zeroes example. In our case, however, there is little risk of that occurring, as:
>
> A) The models are not directly fine-tuned to become more similar to B_i, so should not learn a 'trivial solution'.
>
> B) Even if there could still, theoretically, be a confound where D_fr becomes more "simple"/trivially predictable due to fine-tuning, we believe this is clearly not the case, as the fine-tuned models are able to induce representations which outperform the non-fine-tuned BERT on the targeted-syntactic evaluation tasks.
>
> Furthermore, we have also computed the rank-based metric from Gauthier and Levy which gives the rank of a ground-truth sentence representation in the list of nearest neighbors (computed via cosine similarity) of a predicted sentence representation. We found a strong correspondence between this and the metric we have reported in the paper (which was more stable across subjects, and between datasets), therefore omitted it from the paper for the sake of clarity and space. However, you are correct that including it would offer a more complete picture. We thank you for raising this point. Please find these results for Wehbe 2014 in the table below (we will add this and a similar table for Pereira 2018 to the appendix):
>
> Pre.: pretrained
> Df: domain-finetuned
> Ag: attention-guided finetuning
>
> Wehbe 2014 (Mean and Median ranks are out of a total of 4369 words in dataset):
>
> | Model/Metric     	| Pearson r 	| Mean Rank 	| Median Rank 	|
> |------------------	|-----------	|-----------	|-------------	|
> | Pre.             	| 0.225     	| 436.70    	| 53.13       	|
> |------------------	|-----------	|-----------	|-------------	|
> | Df-baseline-dm   	| 0.204     	| 493.11    	| 89.32       	|
> | Df-baseline-ud   	| 0.206     	| 497.24    	| 81.69       	|
> | Df-baseline-ucca 	| 0.164     	| 689.89    	| 227.30      	|
> |------------------	|-----------	|-----------	|-------------	|
> | Ag-dm            	| 0.343     	| 172.45    	| 10.96       	|
> | Ag-ud            	| 0.280     	| 255.127   	| 18.28       	|
> | Ag-ucca          	| 0.261     	| 315.73    	| 25.78       	|
>
> The table shows that the models which have higher Pearson r scores, also have a lower average ground truth word/sentence nearest neighbour rank i.e. induce representations that better support contrasts between sentences/words which are relevant to the brain recordings. We hope that this adequately addresses your unease re. the methodology of evaluation.
>
> Regarding the first point, we would like to respectfully dispute the characterization of the work’s contribution as incremental: A) we present a novel approach which allows for targeted evaluation of particular structural hypotheses from linguistic theory regarding the composition of meaning in the brain; B) utilising this, we conduct a carefully controlled evaluation involving three different syntactic and semantic linguistic formalisms across two fMRI datasets of different granularities; C) we then present an analysis of a variety of factors including textual domain, ability to model different syntactic constructions, and word class (content vs. function).
>
> Naturally, we agree that a deeper analysis is of interest. The scope of our analysis is necessarily restricted both by space and the amount of information one can reasonably include in an already packed work. We are currently conducting a fine grained analysis of the fine-tuned and non-fintuned models’ representation of semantic information, as suggested by Reviewer #1, and will include it.

---

### Author Response · Authors · 2020-11-24
**Summary of revision**

Dear reviewers, we appreciate your feedback. A lot of minor changes have been added based on your suggestions; we hope we have addressed all of your concerns. To clarify what has been changed, we have compiled the following overview.

 - Based on reviewers’ suggestions, we elaborate on our discussion in section 5, including an analysis of the impact of structural bias on how models encode semantic information. We find that the structurally biased models improve in their ability to make a range of semantic distinctions. Details of how this analysis is conducted and further discussion are added to Appendix E.
- We correct the method by which perplexity is calculated for the analysis of the effect of domain, and include the results for both the domain-fintuned baselines and the structurally biased models in Appendix D. Our main conclusions regarding the effect of domain remain largely unchanged.
- We also correct part of the methodology for our statistical testing as suggested by reviewer #3, and make some adjustments to our discussion of the Pereira 2018 results based on this. The Wehbe 2014 results are not affected by this, due to being highly significant. We also include the details of our statistical testing procedure and results, including testing for strength of generalization across subjects in Appendix C.
- We include the rank-based metrics from Gauthier and Levy, 2018 which gives the rank of a ground-truth sentence representation in the list of nearest neighbors (computed via cosine similarity) of a predicted sentence representation (Appendix B). Alongside the GA models’ improved performance on the targeted syntactic evaluation and the semantic tagging tasks, this provides ample evidence that the main evaluation metric is not ‘trivially gamed’ by the representations induced by the structurally biased models.

---

### Decision · Program_Chairs · 2021-01-07
**Final Decision**

**Decision:**

Reject

**Comment:**

This paper explores the effect on decoding accuracy (predicting hidden representations from fMRI datasets) from fine tuning models by injecting structural bias.  This paper specifically focuses the attention of BERT on syntactic features of the text, which (for one dataset) appears to improve the decoding performance.  The paper's motivation is strong, and complex concepts are communicated clearly.

The review period was very productive.  There were some questions about analyses, and the validity of the statistical tests, but through some very thorough back and forth with the reviewers, this seems to have been resolved.  There is a good amount of analysis done on the resulting language models to try and determine the impact of finetuning or attention on the models. However, the results on the fMRI two datasets appear to be very different, and it's unclear why (and isn't clearly related back to the extensive language model analyses).  We would have liked to have seen a more thorough analysis of the stark difference in performance, and some convincing explanations for the difference based on the analyses.


P.s. A minor point, but the Wehbe paper uses Chapter 9 of Harry potter, not chapter 2.